# A Targeted Metagenomics Approach to Study the Diversity of Norovirus GII in Shellfish Implicated in Outbreaks

**DOI:** 10.3390/v12090978

**Published:** 2020-09-03

**Authors:** Marion Desdouits, Candice Wacrenier, Joanna Ollivier, Julien Schaeffer, Françoise S. Le Guyader

**Affiliations:** Laboratoire de Microbiologie (LSEM), Ifremer, rue de l’Ile d’Yeu, B.P. 21105, CEDEX 03, 44311 Nantes, France; candice.wacrenier@ifremer.fr (C.W.); joanna.ollivier@ifremer.fr (J.O.); julien.schaeffer@ifremer.fr (J.S.); soizick.le.guyader@ifremer.fr (F.S.L.G.)

**Keywords:** norovirus, foodborne virus, metagenomics, amplicon deep sequencing, viral diversity

## Abstract

Human noroviruses (NoV) cause epidemics of acute gastroenteritis (AGE) worldwide and can be transmitted through consumption of contaminated foods. Fresh products such as shellfish can be contaminated by human sewage during production, which results in the presence of multiple virus strains, at very low concentrations. Here, we tested a targeted metagenomics approach by deep-sequencing PCR amplicons of the capsid (VP1) and polymerase (RdRp) viral genes, on a set of artificial samples and on shellfish samples associated to AGE outbreaks, to evaluate its advantages and limitations in the identification of strains from the NoV genogroup (G) II. Using artificial samples, the method allowed the sequencing of most strains, but not all, and displayed variability between replicates especially with lower viral concentrations. Using shellfish samples, targeted metagenomics was compared to Sanger-sequencing of cloned amplicons and was able to identify a higher diversity of NoV GII and GIV strains. It allowed phylogenetic analyses of VP1 sequences and the identification, in most samples, of GII.17[P17] strains, also identified in related clinical samples. Despite several limitations, combining RdRp- and VP1-targeted metagenomics is a sensitive approach allowing the study NoV diversity in low-contaminated foods and the identification of NoV strains implicated in outbreaks.

## 1. Introduction

Human noroviruses (NoV) are the leading cause of acute gastroenteritis (AGE) in the human population (reviewed in 1). They are responsible for epidemics of diarrhea and vomiting in people of all ages worldwide [1]. These small, non-enveloped viruses belong to the family of *Caliciviridae*. Their transmission is fecal-oral, mostly directly from person-to-person, but also through the consumption of contaminated foods or waters [2]. Among these, fresh produces (such as salads or berries) and shellfish are the most frequent foodstuffs associated to NoV foodborne outbreaks [3,4]. NoV replicate in the digestive tract of infected people (either sick or asymptomatic carriers), who shed high amounts of viral particles in their feces [5]. These particles are present in sewage, and discharged from wastewater treatment plants, especially following heavy rainfall and overflows [6]. They are highly resistant in the environment, where they can persist for weeks [7]. Their presence in coastal waters can result in the contamination of filter-feeding bivalve molluscan shellfish [8], especially oysters which express a NoV-specific ligand [9].

NoV have a non-segmented, single-stranded, positive-sense RNA genome of ~7500 nucleotides (nt) containing three open reading frames (ORF) [1]. The NoV genus presents a wide genetic diversity, being divided into at least 10 genogroups (G), among which GI, GII, GIV, GVIII and GIX can infect humans [10]. Genogroups are further divided into 49 capsid (VP1) genotypes and 60 RNA-dependent RNA Polymerase (RdRp) P-types. Indeed, recombination of NoV genomes occurs frequently at the junction between these two genes [11] and the identification of a NoV strain requires the typing of both RdRp and VP1 [10]. In the human population, GII.4 strains were dominant until recently, with new variants emerging every 2–3 years [1,12] and classified as subtypes. In 2014 and 2016, new epidemic strains identified as GII.17[P17], GII.4_2012[P16] and GII.2[P16] have emerged, spread worldwide [13,14,15] and now circulate widely together with the latest GII.4 variants, GII.4_2012[P4_2009] and GII.4_2012[P31] [13].

Genotyping of NoV is of importance to follow the emergence of new strains that are often associated with increased epidemic burden, owing to the lack of protective herd immunity [1,12]. Besides, different strains may show differences in pathogenicity and medical requirements [3]. Foodborne epidemics, such as those associated with shellfish, are more often associated with certain viral genotypes (GI, GII.3 and GII.6) than person-to-person transmission (mainly GII.4) [3,8,16]. Environmentally contaminated waters or foods are also more likely to harbor multiple NoV strains because they are contaminated through discharge of sewage containing the high diversity of strains circulating in the human population [16,17,18]. This increases the risk of viral recombination in infected people [11]. For all these reasons, public health protection and management requires studying the diversity of NoV in environmentally contaminated foods such as shellfish. Current genotyping of NoV in shellfish relies on reverse-transcription and PCR amplification of portions of the RdRp and/or VP1 genes, followed by cloning of the amplicons and their sequencing using the Sanger method [19]. This approach is both sensitive and specific and allows for the identification of the most frequent strains in the sample, but is labor- and time-consuming, and often misses less abundant strains. When analyzing shellfish related outbreaks, genotyping of norovirus in stool and oyster samples may show discrepancies [20] which raises the question of the exact role played by the oyster, and could be overcome by deeper analysis of viral diversity.

High-throughput sequencing (HTS) has opened the door for the study of microbial diversity in the environment. Several HTS approaches have now been applied to foodborne viruses, including agnostic and targeted metagenomics, but they still face many challenges before routine implementation in foods [21]. Among these approaches, deep sequencing of NoV VP1 amplicons has been used several times with sewage or shellfish samples [22,23,24]. Here, we aimed at validating this approach and combining VP1 and RdRp genotyping. For this, we first used artificial samples of known composition and concentration, to assess how the technique performs on samples with low NoV concentration and its ability to recover diverse genotypes. Then, we compared this approach to the classical cloning and Sanger sequencing on NoV-GII-positive shellfish samples associated to confirmed AGE outbreaks, characterized by a low level of NoV contamination.

## 2. Material and Methods

### 2.1. Shellfish Samples

Among samples received in the laboratory between January and May 2016 for suspicion of NoV contamination in the frame of epidemiological investigations following NoV outbreaks in consumers, we selected 10 samples with a mean Ct value for NoV GII below 36 after triplicate qualitative qRT-PCR analysis (see below), and with enough available material for our study. These 10 samples (one clam, two mussel and seven oysters samples) were linked to eight NoV outbreaks involving a total of 26 human cases (Table 1). For two outbreaks, matched stool samples from patients were analyzed and NoV GII.17 was detected (data from the National Reference Center for gastroenteritis viruses, Dijon, France). These samples were not available for other outbreaks.

### 2.2. Nucleic Acids Extraction

Upon reception, shellfishes were shucked, weighed, and their digestive tissues (DTs) dissected as previously described [25]. As an extraction process control, Mengovirus (MgV) (2.10^6^ RNA copies) was added to each DT sample (2 g) before incubation with 2 mL of proteinase K solution, following the ISO norm for NoV detection in foods (ISO 15216–1:2017). The NucliSens extraction kit (BioMérieux, Marcy-l’Étoile, France) was used to recover nucleic acids (NA) from 0,5 mL of DT supernatant or MgV control, following the manufacturer’s recommendations, with a final elution step of 5 min at 60 °C. The eluted nucleic acids were directly used for NoV GII and MgV genome detection and further stored at −20 °C.

### 2.3. Detection of Viral Genomes by rRT-PCR

Primers and probes previously described were used for MgV and NoV detection by real-time RT-PCR (ISO 15216–1:2017). Specifically, NoV GII was detected using QNIF2 and COG2R primers, and QNIFs probe [26,27]. The rRT-PCR was carried out with the Ultrasense One-step quantitative RT-PCR system (Fisher Scientific, Illkirch, France) using 5 μL of NA per well (final volume of 25 μL) on a Mx3000P QPCR System (Agilent Technologies, Les Ulis, France). Undiluted and 10^-1^ dilutions of NA were processed in triplicates. Negative controls (H_2_O and negative DT extract) and a positive control based on in vitro transcription of plasmids containing nucleotides 4191–5863 of the GII.4 Houston virus (Genbank EU310927) were included in each run. The cycle threshold (Ct) value enabling quantification was defined as the cycle that showed a significant increase in fluorescence. Filter tips and dedicated rooms for mix preparation and NA handling were used to prevent false positives.

### 2.4. Extraction Efficiency and Inhibition

After extraction of shellfish samples seeded with process control virus, MgV genome quantification was performed by rRT-PCR to determine the extraction efficiency, expressed as a percentage of virus recuperation for each sample and to verify the absence of inhibitors [25]. Extraction efficiencies were above 1% for all samples, and no inhibition was observed.

### 2.5. Artificial Samples of Known Composition

To assess some of the biases induced by targeted metagenomics, we built artificial samples of known composition and concentration by pooling several NoV strains. NA from 10 NoV GII-positive stool samples of known RdRp and VP1 genotypes were quantified for NoV GII by rRT-PCR as described above, using 10^-1^ and 10^-2^ dilutions of NA in water, each in triplicates. A negative control (H_2_O) was included, and quantification was performed using a 6-point standard curve (in duplicates) based on the NoV GII.4 control described above. NA from stools were adjusted to 10^3^ NoV GII genome copies (gc)/μL, and mixed into 4 pools (A, B, C and D) with different distributions. Each pool was diluted to reach a final total concentration of 10^2^ or 10 NoV GII gc/μL, generating 8 artificial samples (Table 2).

We recently used the Digital QuantStudio 3D Digital PCR system (Fisher Scientific, Illkirch, France) to detect NoV GII in shellfish by two-step RT-PCR [28]. Here, to accurately quantify the concentration of NoV GII in the artificial samples (Table 2), this protocol was adapted to conduct one-step RT-PCR, using the same primers and probes as for rRT-PCR, the Ultrasens One-step kit (Fisher Scientific), GE sample loading reagent (Fluidigm, Les Ulis, France) to optimize the viscosity, and 3 µL of NA per chip.

### 2.6. Generation of RdRp and VP1 Amplicons

RdRp and VP1 amplicons were generated from undiluted NA from BMS samples (stored at −20 °C for 6 months before analysis) and from each artificial sample. 5 μL of viral RNA were retro-transcribed using the SuperScript III kit (Life Technologies, Villebon sur Yvette, France) using the manufacturer’s instruction with primers specific for RdRp (P110, 5’-ACDATYTCATCATCACCATA-3’) [29] or VP1 (GIISKR, 5’-CCRCCNGCATRHCCRTTRTACAT-3’) [27]. Viral cDNA were then submitted to a first PCR using the Platinum Taq kit (Life Technologies, France), with additional forward primer for RdRp (NV4611, 5’-CWGCAGCMCTDGAAATCATGG-3’) [30] or VP1 (QNIF2D, 5’-ATGTTCAGRTGGATGAGRTTCTCWGA-3’) [26], respectively, and 40 amplification cycles. These first PCR products were used for a ½ nested PCR using the same conditions and an internal forward primer for RdRp (NV4692, 5’-GTGTGRTKGATGTGGGTGACTT-3’) [30] or VP1 (GIISKF, 5’-CNTGGGAGGGCGATCGCAA-3’) [27] and the reverse primers used for RT. Each series of PCR comprised a negative control with water in place of the NA, and the product of this control for the 1^st^ PCR was also used as a template for the ½ nested PCR. A volume of 10 μL of the final PCR products were loaded on a 9% acrylamide gel (Protogel, National Diagnostics, Nottingham, United Kingdom) in TBE buffer with the MassRuler DNA loading dye (Fisher Scientific) and the DNA molecular weight marker VIII (Roche Diagnostics, Meylan, France). After migration gels were incubated in TBE with 0.01% of GelRed (Biotium, Fremont, CA, USA) and visualized using a ChemiDoc imaging system and the Quantity One software (Bio-Rad, Marnes-la-Coquette, France). Final RdRp and VP1 amplicons were 193 bp and 344 bp long, respectively. No PCR products were observed in the negative controls.

### 2.7. Classical Genotyping: Cloning and Sanger Sequencing

RdRp and VP1 amplicons from shellfish were used fresh or after storage at −20 °C for a few months. They were cleaned using the QIAquick PCR purification kit (Qiagen, Courtaboeuf, France) and cloned using the TOPO-TA cloning kit for PCR products (Fisher Scientific), following the manufacturers’ instructions. For each sample and gene, 8 colonies with insert (white color) were picked, screened by on-colony-PCR and grown in LB with 100 μg/mL of ampicillin for 16–24 h at 37 °C. For on-colony-PCR, each colony was dipped into the PCR mix (Platinum Taq kit, Fisher Scientific) with the same primers as the ½ nested PCR and submitted to the same PCR program. A volume of 10 μL of PCR products were analyzed by poly-acrylamide gel electrophoresis as described above. All bacterial cultures presenting a band at the expected size were used for plasmid extraction using the QIAPrep Spin Miniprep kit (Qiagen) following the manufacturer’s instructions. Plasmids were Sanger-sequenced in both directions using the M13F and M13R primers. Chromatograms were analyzed using Geneious 11.1.4 (BioMatters Ltd., Auckland, New Zealand), their ends were trimmed from bases with error probability above 5%, and contigs were built from forward and reverse sequences by *de novo* assembly. The consensus sequences were genotyped using the Norovirus Typing Tool 2.0 [31]. Unique VP1 and RdRp sequences from each sample are accessible in GenBank under accession numbers MT582385-MT582411 (VP1) and MT586039-MT586064 (RdRp).

### 2.8. Targeted Metagenomics: Amplicon Tagging and Illumina Sequencing

NA concentrations in semi-nested PCR products were quantified after storage at −20 °C for 6–12 months, using High Sensitivity DNA chips and reagents on an Agilent 2100 Bioanalyzer System (Agilent Technologies, France) or the Qubit dsDNA BR assay kit on a Qubit 3.0 (Fisher Scientific). Libraries were prepared from 15 μL of the PCR products using the NEB Next Ultra II DNA library prep kit for Illumina (New England Biolabs, Evry, France) following the manufacturer’s instructions. One negative control for each gene was processed in parallel with samples. Amplicons were size-selected using Agencourt AMPure XP beads (Beckman Coulter, Villepinte, France) with beads/library ratios recommended for insert sizes of 300–400 bp for VP1, and 200 bp for RdRp. NEB Next Index Primers Set 1, 2 and 4 (New England Biolabs) were used to label amplicons during an 8-cycles enrichment PCR. Tagged amplicons were cleaned using AMPure XP beads, quantified on a Qubit 3.0 and analyzed on an Agilent 2100 Bioanalyzer as described above, to check the final size and repartition of DNA segments. They were pooled, mixed with 20% PhiX library, and sequenced paired-end on a MiSeq (Illumina, Paris, France) using the MiSeq Reagent kit v2–600 cycles (2 × 300) for all artificial samples and for VP1 from shellfish samples, and 300 cycles (2 × 150) for RdRp from shellfish samples. Raw reads from each run are accessible in the Sextant repository (https://doi.org/10.12770/6e0c6687-22fe-47e0-bc67-81dde633d25b).

### 2.9. Bioinformatics and Statistics

#### 2.9.1. Quality Trimming

Demultiplexed fastq generated by the MiSeq were analyzed using the FastQC (v0.71) and MultiQC (v1.5.0) tools implemented in Galaxy 2 [32]. The Galaxy tool Trimmomatic (v.0.36.4) was used to remove remaining Illumina indexes, low-quality bases with Phred score ≤ 28 at the 3′ end and resulting reads shorter than 50 bases. A total of 300b reads obtained for RdRp libraries from artificial samples were also cut at 190b using Trimmomatic. Trimmed fastq files were then processed separately for RdRp and VP1, using two different approaches.

#### 2.9.2. Clustering

The FROGS suite (v2.0.0) [33] was used to generate clusters of nearly-identical reads, with the following parameters for preprocess (mismatch rate 0.1, size between 300 and 400 bp for VP1, and 150 and 250 bp for RdRp) and clustering swarm (distance of 3 and de-noising step). Chimeric clusters were removed using FROGS remove chimera. Finally, FROGS filters were used to keep only clusters with an abundance ≥ 0.02%. Fasta sequences of these clusters were uploaded in the Norovirus Typing tool 2.0 for genotype assignation, and further analyzed using Geneious v11.1.4 (BioMatters Ltd.) and AliView 1.18 [34]. Remaining chimeric clusters exhibiting nucleotide signatures of two or more different genotypes, as well as the few clusters not assigned to NoV, represented 0.11% and 0.15% of the final reads for VP1 and RdRp respectively, and were removed from analysis. To remove noise, a minimal number of 100 reads in a replicate was also required to consider the cluster as present. The abundance of remaining genotypes in each cluster was calculated for each replicate.

#### 2.9.3. Mapping

Trimmed reads were mapped using Bowtie2 (Galaxy tool v.2.3.4.3) on a set of NoV VP1 or RdRp sequences (Appendix A) belonging to each GII genotype and P-type according to the recent norovirus classification [10]. The proportion of reads mapped on each genotype was calculated and genotypes were considered present in the samples when accounting for more than 0.02% of the final read abundance. A minimal number of 100 reads mapped on a reference was also required to consider the genotype detected in each replicate.

#### 2.9.4. Analysis of Sequences and Phylogeny

Using Muscle (v3.8.425) implemented in Geneious, sequences of RdRp and VP1 clones and clusters from shellfish samples were aligned together and compared. VP1 unique clones and clusters were aligned to reference NoV GII, GIII and GIV sequences from the Norovirus Typing Tool v2.0, and to their best hits of a blastn search on the nucleotide database of NCBI. A Bayesian phylogenetic tree was built using Mr Bayes (v.2.3.6) with the GTR evolutionary model, 2 million generations, a sampling frequency of 1/400 and 20% burn-in.

### 2.10. Statistics

To compare the observed NoV GII genotypes to the theoretical composition of the different pools, chi-square tests were performed using GraphPad Prism version 6.00 for Windows (GraphPad Software, San Diego CA USA).

## 3. Results

### 3.1. Assessing the Targeted Metagenomics Approach with Samples of Known Composition

Deep sequencing of PCR amplicons is a promising approach to identify NoV genotypes contaminating environmental samples, but it may induce several biases, such as over- or under-representation of a certain genotypes, or sampling bias especially when the viral concentration is very low. Besides, the reproducibility of the method remains unknown. To assess this, 8 artificial complex samples of known composition were prepared by pooling NA from stool samples, to obtain 4 different compositions (A, B, C, and D), each at two dilutions (100 and 10 gc/µL) (Table 2). The achieved concentration as measured in each pool of NA by digital PCR was comprised between 90 and 150% of the target concentration, except for pool D10, which was 4 times more concentrated than intended (Table 2).

From each pool, two portions of the NoV GII genome, in the RdRp region (positions 4440 to 4591) and the VP1 region (positions 5064 to 5366), were amplified in triplicate. RdRp amplification yielded PCR products of the expected size for all pools and replicates, except one replicate of the B10 pool. For VP1, the three replicates were obtained for 6 pools, and two replicates for pools B10 and C10. All PCR products were deep-sequenced and yielded between 136.278 and 705.006 raw reads (mean 234.174 reads for VP1, 459.501 for RdRp). Negative controls were included in the analysis and yielded 40.315 and 65.614 raw reads for VP1 and RdRp, respectively.

After quality trimming, reads were subjected to two methods of analysis. In the first one, they were clustered into consensus sequences that were genotyped using the online Norovirus Typing Tool 2.0. Thresholds were set to detect the expected P-types and genotypes with good confidence, which excluded clusters that accounted for 0.6 and 0.3% of the reads for RdRp and VP1, respectively. As a result, the few reads obtained for the negative controls (water) were filtered out, and no NoV GII were identified in these samples. In the 8 pools, we detected 6 of the 9 expected P-types: GII.P17, GII.P21, GII.P4_2009, GII.P31, GII.P33 and GII.PNA7. Each was represented by one cluster except GII.P33, which was represented by 3 clusters. Three expected P-types (GII.P2, GII.P16 and GII.P7) were not detected. In addition, GII.P17 was detected with a very low abundance in one D100 replicate devoid of GII.17[P17]. For VP1, one A10, one B10 and the three C100 replicates did not pass the thresholds set for NoV identification. With the remaining pools and replicates, we obtained 9 clusters that were assigned to 7 of the 8 expected genotypes: GII.17, GII.2, GII.3, GII.4_2012, GII.6, GII.1 and GII.12. Only GII.7 was not detected. There was one cluster per genotype, except GII.2 and GII.4, which were both represented by two clusters. Interestingly, this reflected the initial composition of artificial samples in which two GII.2 and two GII.4 strains were mixed. Comparison of the cluster’s sequence with the available sequences of each strain confirmed that each cluster corresponded to one strain with 97.13 to 100% identity (Table 3).

In the second method, trimmed reads were directly mapped to a set of 57 RdRp or 46 VP1 references sequences, one per GII P-type and genotype, respectively (Appendix A). To allow comparisons, the detection thresholds defined for the clustering methods were also applied for the mapping. This resulted in filtering out all reads from the negative controls but also from one A10 and the three C100 replicates for VP1. Importantly, the same expected P-types and genotypes were identified with this analysis as with the clustering. For RdRp, GII.P41 was also detected, which is compatible with the RdRp of strain S596 present in the pools (Table 2). GII.P17 was identified in three D100 and two D10 replicate where this genotype was not present, suggesting a contamination. Additional, false-positive P-types (GII.P1 and several GII.P4 subtypes), and genotypes (GII.10 and GII.4_2007EU) were detected with a very low final abundance comprised between 0.024 and 0.30%. As with the clustering analysis, three expected P-types (GII.P2, GII.P16, GII.P7) and one expected genotype (GII.7) were not mapped by any read.

For each pool, the proportion of reads assigned to each genotype was plotted as the mean of the three replicates, for the two methods separately (Figure 1). Overall, the distribution of genotypes observed in the different pools was very similar between mapping and clustering analysis methods. All observed distributions differed from the expected ones (clustering results, chi2 test, *p* < 0.0001). Differences in the detection or proportion of P-types and genotypes were observed between replicates, especially when the expected proportions were low, such as in the B and C distributions for genotypes others than GII.17[P17]. Yet the technique was able to clearly identify the main strain (GII.17[P17]) in these pools.

### 3.2. Diversity of NoV GII in Outbreak-Related Shellfish Samples

During winter 2016, the emerging GII.17[P17] strain was reported in oysters from France [19]. To further document the implication of this new strain in shellfish-borne NoV outbreaks, we selected 10 shellfish samples linked to 8 different outbreaks, consisting of mussels and clams imported from Spain, and oysters from different production areas in France (Table 1). Following semi-nested PCR, amplicons were obtained for all samples but C3988 for RdRp and C4009 for VP1, cloned and sequenced. We obtained 59 GII RdRp clones and 32 VP1 clones. We identified 3 P-types (GII.P17, GII.P21 and GII.P31) and 4 genotypes (GII.3, GII.4, GII.13 and GII.17) (Table 3). In the three samples from Spain, we observed the presence of diverse NoV GII genotypes, either for the RdRp (C3994) or VP1 (C3988) genes, or for both (C4006). In samples from France, only GII.17 and GII.P17 were detected (Table 4).

To gain further insight into the diversity of NoV GII in these shellfish samples and validate the deep-sequencing approach in contaminated foods, we sequenced the leftovers of purified semi-nested PCR amplicons as described for artificial nucleic acid pools. We obtained 509.973 to 994.285 reads for RdRp (mean 678.661) and 987 to 1.098.178 reads for VP1 (mean 702.213), which were quality-trimmed and subjected to analysis by clustering as described earlier. RdRp analysis yielded 12 clusters that were assigned to NoV GII P-types GII.P17, GII.P4, GII.P21, GII.P31 and GII.P33 (Table 3). For VP1, we obtained 23 clusters assigned to NoV GII.17, GII.13, GII.1, GII.4, GII.2, GII.3, GII.6, GII.14 and to NoV GIV (Table 4).

Overall, 8 samples had diverse NoV genotypes, the three samples from Spain and five from France. Figure 2 shows the relative abundance of the different clusters in each sample. In agreement with the results obtained through cloning, GII.17 and GII.P17 were the most abundant in all samples but C4006 (Figure 2). In two samples linked to the same outbreak (C4008 and C4009), only GII.17 and P17 were detected. Two other samples linked to another outbreak (C4019 and C4020) shared a predominance for GII.17 and P17 but with diverging minority genotypes (GII.4 in one case and GII.13 in the other) (Figure 2).

### 3.3. Analysis of NoV GII Sequences from Shellfish Samples

To understand better the intra-genotype diversity of strains detected here, sequences of RdRp and VP1 clusters and clones were aligned and compared. For GII.P17, the different sequences were very similar with percent identities between 97.33 and 100%. There were 36 clones, belonging to the different samples, actually identical to the most abundant cluster, found in all samples. Another group of 4 identical P17 clones was shared between C4013 and C4019. For P21 and P31, one pair of identical cluster and clone was observed among the different sequences, but percent identities tended to be lower between the other sequences, ranging from 88.08% to 100% and from 89.40% to 100%, respectively.

For VP1, percent identities among GII.17 sequences were high, ranging from 96.69% to 100%. There were 11 identical GII.17 clones obtained from 6 samples, also identical to the most abundant VP1 cluster, which was found in all samples. All GII.13 sequences (2 clones, and the second most abundant cluster) were identical. Percent identities among the GII.4 sequences were also high, ranging from 96.35% to 100%, one clone being identical to a cluster. For GII.2, they ranged from 96.69% to 98.34%. For the other clusters, they were lower, ranging from 91.39% to 97.68% for GII.6, 90.73 to 99.67% for GII.3, and being of 95.70% between the two GII.14 clusters. Following Bayesian phylogenetic analysis, VP1 unique clones and clusters segregated together with reference strains of the same genotypes as expected, in clades supported by high posterior probability (Figure 3). For GII.3 and GII.6, sequences from shellfish were robustly split in different clades within their genotype, supporting co-circulation of different strains contaminating shellfish. For GII.17, the many clusters and clones were more closely related to the strains that emerged in 2014 and 2015 (Kawasaki_323 and Kawasaki_308) than to the more ancient GII.17 references (Figure 3). Together, these results show that an emerging GII.17[P17], or a group of very similar strains, was contaminating shellfish in Europe in early 2016, together with a diversity of other strains circulating in the population, leading to some AGE cases.

## 4. Discussion

Analysis of microbial diversity in environmental and food samples through metabarcoding is a widely used approach relying on the deep sequencing of specific genomic regions such as the 16S rRNA gene for bacteria. Given the high genetic variability of viruses and their lack of common ancestry, a pan-virus metabarcoding/targeted metagenomics approach is not possible. Some studies have applied this approach to a specific virus [24,35,36] or viral families sharing a conserved gene such as T4-like DNA bacteriophages or picorna-like RNA viruses [37]. Here, we aimed at describing the diversity of norovirus GII in contaminated food samples by combining VP1 and RdRp targeted metagenomics, and first assessed the technique on artificial samples.

### 4.1. Advantages and Limitations of the Deep-Sequencing Approach

#### Comparison of Methods for Bioinformatics Analysis

To analyze our data, we chose to compare two different bioinformatics methods, both allowing to identify the genotypes present in the sample, and to calculate the relative abundancy of each genotype per sample. The first one, clustering, took advantage of a bioinformatics pipeline optimized for 16S/18S metabarcoding, the FROGS suite, which is operated in the user-friendly interface of a Galaxy server [33]. The second one, mapping, was adapted from [24] on NoV GII VP1 deep sequencing in oyster samples. Both gave very similar results, but the clustering had several advantages. It detected only the expected genotypes in the pools, none in the negative control, and only one replicate appeared contaminated with GII.P17 reads in pools where it should be absent. In contrast, mapping appeared less stringent, several additional genotypes being detected, albeit at a very low level, and GII.P17 contaminating most replicates in the pools devoid of this strain. Results obtained with mapping may also be dependent on the set of chosen reference sequences [38]. Finally, the generation of clusters with a consensus sequence allowed re-sequencing of the strains, with nearly identical sequences even after the two successive PCR needed for food samples (Table 3). This allowed to identify the two pairs of different strains belonging to the same genotypes (GII.2 and GII.4) and can be useful for phylogenetic analyses. Therefore, clustering was chosen for analysis of deep-sequencing data from shellfish samples.

### 4.2. Impact of NoV Concentration

The concentration of NoV in environmentally contaminated foods such as shellfish is very low. To validate our method, we set the total NoV GII concentration of the artificial samples at low concentrations (100 and 10 gc/µL in NA extract), which is still one order of magnitude higher than average shellfish samples [28,39]. We observed a high impact of virus concentration, with failure to amplify VP1 and/or RdRp from several replicates and under-detection of some NoV strains in the samples with the lowest concentration of NoV, especially for minority strains. Since only a small portion of NA extract is used for this method, loss of diversity and increase of variability in samples with low concentration were expected. Nonetheless, when applied on actual shellfish samples with even lower NoV GII levels, the deep-sequencing approach was still able to explore the diversity of genotypes and P-types, although this diversity may be under-estimated. Considering these promising results, increasing the portion of NA extract used for this approach may contribute to lower these biases. Here, although we estimated the impact of NoV concentration, we did not fully assess the sensitivity of the technique. It is likely to be similar to that of classical cloning and sequencing, since it depends on the generation of a PCR product and could be tested on shellfish samples with even higher NoV GII Ct values.

### 4.3. Selectivity

The artificial samples were used to assess if our approach could describe accurately the diversity of NoV GII. Here, it failed to detect 3 P-types (GII.P7, P2 and P16), and one genotype (GII.7), independently of the method used for bioinformatics analysis (Figure 1). It also repeatedly over-estimated the proportion of some genotypes and P-types (Figure 1). Although we cannot exclude a role of the sequencing platform, the selection of genotypes and distortions in their relative abundance are likely due to the use of degenerate primers and multi-template PCR, as observed for 16S bacteria metabarcoding [40]. Indeed, P-types or genotypes missed in artificial samples were also not observed in shellfish samples using cloning or targeted metagenomics. We chose primers commonly used for the detection and sequencing of NoV GII RdRp and VP1, that were previously found to allow amplification from most NoV GII genotypes, including GII.7 [22,41]. New primer design could help to overcome the limitation we observed, but given the high genetic variability of NoV, conserved genomic regions allowing the design of broad-range primers are scarce. Thus, understanding the biases induced by these primers is important for subsequent evaluation of NoV diversity in environmental samples.

### 4.4. Comparison of Classical Genotyping and Targeted Metagenomics on Shellfish Samples

Overall, the deep sequencing allowed to identify two GII P-types, 4 genotypes and importantly another genogroup (GIV) that were missed by cloning and Sanger sequencing. For each shellfish sample, results obtained with cloning and deep sequencing were compatible, with more diversity with the latter approach, as expected and observed previously on sewage samples [22]. In one case only, a P-type identified by cloning was not detected by deep sequencing (Table 3, sample C3994, RdRp gene). In addition, majority sequences obtained using the two approaches (groups of identical clones, most abundant clusters) were identical or highly similar, which confirms the re-sequencing of strains observed using artificial samples. The quality of the VP1 sequences obtained with both methods (clones and clusters) allowed phylogenetic analyses. For RdRp, they were too short to yield robust results. Finally, deep sequencing required less lab-work than tedious cloning, clone screening and plasmid purification, and the bioinformatics analysis of data was possible in a user-friendly interface. Given these advantages, this approach could be easily implemented in laboratories investigating NoV contamination in foods to gain better insight in the diversity of NoV genotypes and their propensity for foodborne transmission.

### 4.5. NoV Diversity in Shellfish Samples

Both classical cloning and targeted metagenomics confirmed that shellfish samples can harbor diverse NoV strains, although this diversity may be underestimated by the technical biases described above. Most genotypes and P-types detected here were frequently identified in outbreaks or environmental samples in 2016 and before [13]. Others were less common: in 2016, GII.1, GII.13 and GII.14 respectively represented 0.6, 0.7 and 1.1% of NoV VP1 sequences reported in the NoroNet network, and P33, 1.5% of RdRp [13]. All were previously reported in shellfish from different countries [8,24,42]. The detection of GII.13 in 5 different shellfish samples from France and Spain, with high relative abundance, suggests that contaminated shellfish may be an efficient mode of transmission for this otherwise rare genotype. Confirmation would require the analysis of stool sampled during shellfish-related outbreaks. The GIV genogroup was previously reported in contaminated shellfish [43] but is often overlooked due to its rare occurrence and the need for dedicated primers. Interestingly, while GI NoV were not detected here, the broad-range NoV GII VP1 primers allowed the detection of GIV in one shellfish sample. This is not surprising considering the genetic proximity between GII and GIV and suggests that our approach could help identifying GIV in environmentally contaminated foods.

Deep sequencing of VP1 amplicons was described before, but an original point of this study is to combine RdRp and VP1 data. GII.17[P17], GII.4_2012[P31] and GII.3[P21] were known to circulate widely in 2016 [13], which is compatible with the co-detection of GII.17 and GII.P17 in all samples, and of GII.3 - GII.P21 and GII.4 - GII.P31 in sample C4006. In some samples, RdRp and VP1 data do not completely match, most likely because the corresponding genotype or P-type was not detected, or possibly because of the presence of a rare recombinant. This underlines the limit of an approach with separate RdRp and VP1 amplicons, which could be overcome by targeting the RdRp-VP1 junction.

The identification of diverse but very similar sequences for GII.17 and GII.4_2012 (Figure 3) is in line with previous findings in shellfish and may be due to contamination by quasispecies of each strain initially present in sewage [19]. GII.17 and GII.P17 sequences detected in all samples were closer to the emerging Kawasaki strains from 2014 and 2015 (Figure 3). This confirms previous reports that this GII.17[P17] variant contaminated the environment and shellfish during winter 2015–2016 when it spread worldwide [19,44,45,46]. Importantly, for two outbreaks, GII.17 was identified in the stools of patients (Table 1), and, using our approach, in the corresponding shellfish samples (Table 3). Thus, targeted metagenomics in shellfish related to NoV outbreaks can help to link AGE cases definitely to the contaminated foodstuff.

In conclusion, targeted metagenomics induced some biases, as revealed by the use of artificial samples with multiple NoV strains but could identify most of these strains. On shellfish samples, the classical cloning/sequencing approach and the deep sequencing of amplicons gave compatible results, and the latter approach revealed a higher diversity in NoV genotypes and P-types. Some outbreak-related shellfish samples exhibited a high diversity of NoV both with VP1 and RdRp regions, and most were dominated by GII.17[P17]. Our results show that this approach can be applied successfully to food samples contaminated with low levels of NoV GII and could be implemented in laboratories for the analysis of NoV diversity.

## Figures and Tables

**Figure 1 viruses-12-00978-f001:**
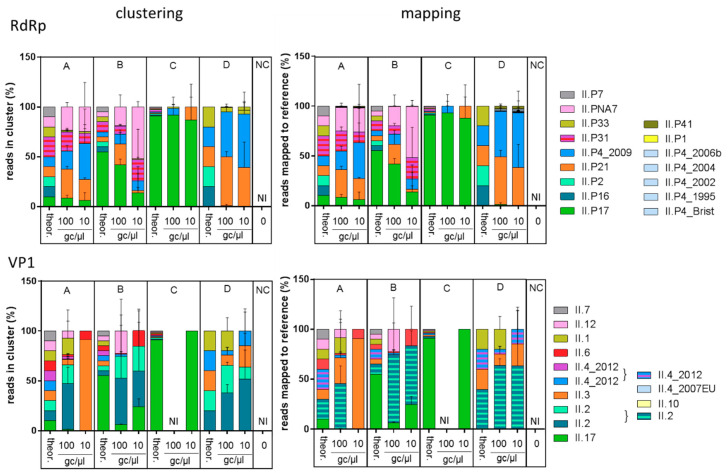
Relative abundance of NoV GII P-types and genotypes identified in the artificial samples. For each nucleic acid pool (A, B, C, D) and dilution (100 and 10 gc/µl), and the negative controls (NC), the proportion of reads assigned to a given genotype is depicted as the mean of the replicates with the standard deviation as error bars, for clustering (left) or mapping (right) analysis of RdRp (top) and VP1 (bottom). The theoretical composition is also depicted for comparison. Colors are matched between RdRp and VP1 graphs according to the NoV strains that were used to generate the samples (example: GII.12[PNA7] = S23 = pink). When unable to differentiate two strains with the same RdRp (P31) or genotype (GII.2, GII.4), both colors were used as stripes. NI: not identified.

**Figure 2 viruses-12-00978-f002:**
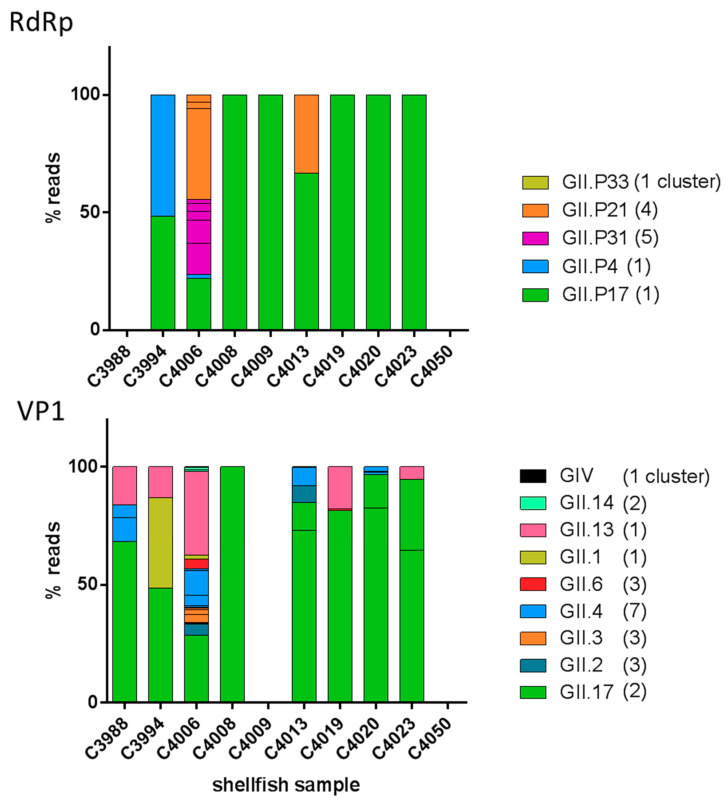
Relative abundance of NoV GII P-types and genotypes identified in the shellfish samples. The proportion of reads belonging to the clusters obtained for RNA-dependent RNA Polymerase (RdRp) (top) and VP1 (bottom) are depicted with the color corresponding to their genotype. The number of clusters obtained per genotype is indicated between parentheses.

**Figure 3 viruses-12-00978-f003:**
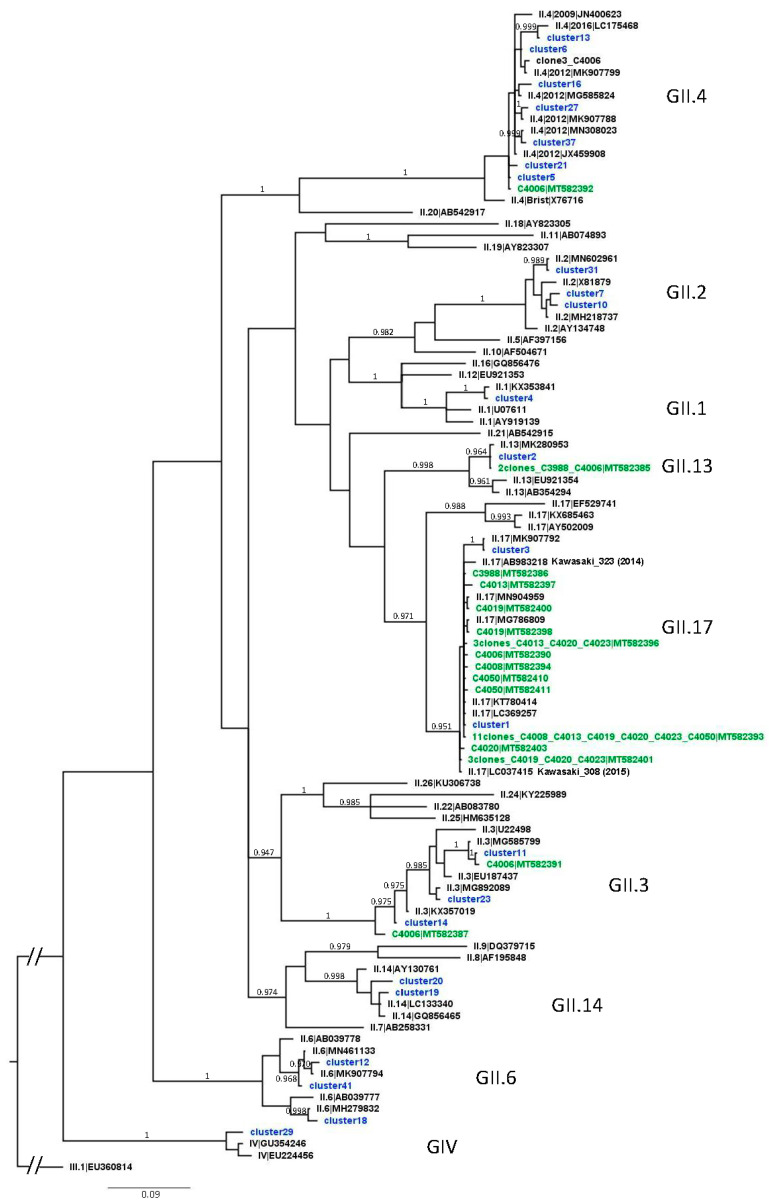
Phylogenetic tree of VP1 clones and clusters from shellfish samples. Bayesian phylogenetic analysis was used to generate this tree with NoV GII, GIII, and GIV sequences retrieved from GenBank (black, with genotype|accession number), sequences of clones (green, with sample name|accession number), and of clusters (blue) obtained from the shellfish samples. Posterior probabilities above 0.95 are indicated just above or below the supported branch. The GIII.1 sequence was used as outgroup to root the tree.

**Table 1 viruses-12-00978-t001:** List and characteristics of outbreak-related shellfish samples.

Sample	Shellfish Species	Sampling Date	Country of Origin/Coastal Area	Cases/Exposed	NoV in Patients Stool	Mean NoV GII Ct Value
**C3988**	Mussel	20/01/2016	Spain	2/2	GI.1, GII.17	32
**C3994**	Mussel	28/01/2016	Spain	2/10	GII.17	35
**C4006**	Clam	24/02/2016	Spain	2/3	NA	35
**C4008**	Oyster	01/03/2016	France/Channel	4/13	NA	36
**C4009**	Oyster					35
**C4013**	Oyster	02/03/2016	France/Atlantic	5/8	NA	36
**C4019**	Oyster	26/02/2016	France/Atlantic	4/4	NA	36
**C4020**	Oyster					36
**C4023**	Oyster	10/03/2016	France/Atlantic	4/5	NA	36
**C4050**	Oyster	04/04/2016	France/Atlantic	3/4	NA	34

NA: not available.

**Table 2 viruses-12-00978-t002:** Proportion of human noroviruses (NoV) strains in artificial samples.

Strain	Year	GII. Genotype		Proportion of NoV Strain in Artificial Samples							
		VP1	RdRp	A100	A10	B100	B10	C100	C10	D100	D10
S582	2016	17	[P17]	10%		55%		91%		-	
S587	2009	3	[P21]	10%		5%		1%		20%	
S596	2017	1	[P41/P33] *	10%		5%		1%		20%	
S597	2017	4_2012 ^a^	[P4_2009]	10%		5%		1%		20%	
S598	2017	2 ^b^	[P16]	10%		5%		1%		20%	
S570	2015	4_2012 ^a^	[P31] ^c^	10%		5%		1%		-	
S555	2014	6	[P31] ^c^	10%		5%		1%		-	
S514	2013	2 ^b^	[P2]	10%		5%		1%		20%	
S510	2012	7	[P7]	10%		5%		1%		-	
S23	2007	12	[PNA7]	10%		5%		1%		-	
**Total NoV GII concentration, gc/µl**			Target	100	10	100	10	100	10	100	10
			Measured	149	9.08	122	16.3	66.2	18.5	90.6	41.8

* Using the Norovirus Typing Tool 2.0, S596 RdRp sequence (122 bp) falls into a clade grouping [P41] and [P33]. The best hit using Blastn, KF895877.3, is assigned to [P33]. ^a^ S570 and S596 VP1 both belong to II.4_2012 Sydney, sequences differing by 2 nt/181. ^b^ S598 and S514 VP1 both belong to II.2, sequences differing by 4 nt/209. ^c^ S570 and S555 RdRp are assigned to [P31] and the available sequences are identical.

**Table 3 viruses-12-00978-t003:** Sequence similarity between the NoV strains and the clusters obtained by deep sequencing of nucleic acids (NA) pools.

Strain		S570	S597	S598	S514	S510	S287	S555	S582	S596	S23b
GII P-Type		P31	P4	P16	P2	P7	P21	P31	P17	P33/41	PNA7
GII Genotype		4_2012	4_2012	2	2	7	3	6	17	1	12
**RdRp**	Cluster1	72.80	76.92	73.84	79.81	NA	75.50	75.47	100.00	75.82	76.24
	Cluster2	83.20	83.76	79.14	83.65	NA	100.00	83.02	75.50	79.92	87.13
	Cluster3	85.60	100.00	79.14	77.89	NA	82.78	87.74	75.50	78.28	78.22
	Cluster4	79.20	80.34	77.15	79.81	NA	83.44	79.25	76.16	78.28	100.00
	Cluster5	100.00	86.33	78.48	77.89	NA	82.78	100.00	72.85	78.28	85.15
	Cluster6	79.20	81.20	77.82	75.00	NA	80.80	82.08	76.82	97.13	79.21
	Cluster9	80.00	82.05	77.15	77.89	NA	79.47	83.02	77.48	91.39	81.19
	Cluster10	82.40	82.91	78.48	78.85	NA	80.13	83.96	74.17	92.21	79.21
**VP1**	Cluster1	74.17	76.38	100.00	98.09	77.76	79.86	77.18	78.09	79.88	79.58
	Cluster2	74.17	74.17	76.37	74.16	76.94	80.92	77.59	100.00	84.02	82.08
	Cluster3	72.52	74.72	97.89	100.00	76.94	79.86	76.35	76.33	79.29	80.83
	Cluster4	77.48	76.93	78.06	77.03	82.65	99.65	78.01	81.27	81.66	76.25
	Cluster5	71.85	73.62	79.75	79.90	75.71	78.09	79.67	83.04	91.72	100.00
	Cluster6	96.36	99.59	71.31	71.77	77.14	75.97	75.93	75.27	78.11	69.58
	Cluster7	73.18	80.80	76.37	74.64	80.82	77.39	100.00	76.68	81.07	79.17
	Cluster8	73.51	76.38	78.48	78.47	76.94	81.27	78.84	80.92	100.00	86.25
	Cluster9	100.00	98.48	72.15	71.29	77.14	78.09	75.93	75.27	76.92	69.58

**Table 4 viruses-12-00978-t004:** NoV genogroups, genotypes, and P-types detected using cloning and deep-sequencing approaches on shellfish samples.

Gene	RdRp			VP1		
Sample	Amplicon	P-types of Clones (n. of Clones)	P-types Identified by Deep Sequencing	Amplicon	Genotypes of Clones (n. of Clones)	Genogroups/Genotypes Identified by Deep Sequencing
C3988	No	NA	NA	Yes	GII.17 (1), GII.13 (1)	GII.17, GII.4, GII.13
C3994	Yes	GII.P17 (6), GII.P21 (1)	GII.P17, GII.P4	Yes	NA	GII.17, GII.1, GII.13
C4006	Yes	GII.P31 (4), GII.P21 (2), GII.P17 (1)	GII.P17, GII.P4, GII.P21, GII.P31, GII.P33	Yes	GII.3 (2), GII.4 (2), GII.13 (1), GII.17 (1)	GII.17, GII.2, GII.3, GII.4, GII.6, GII.1, GII.13, GII.14, GIV
C4008	Yes	GII.P17 (7)	GII.P17	Yes	GII.17 (2)	GII.17
C4009	Yes	GII.P17 (5)	GII.P17	No	NA	NA
C4013	Yes	GII.P17 (7)	GII.P17, GII.P21	Yes	GII.17 (3)	GII.17, GII.2, GII.4, GII.6
C4019	Yes	GII.P17 (6)	GII.P17	Yes	GII.17 (4)	GII.17, GII.13, GII.6
C4020	Yes	GII.P17 (6)	GII.P17	Yes	GII.17 (4)	GII.17, GII.4
C4023	Yes	GII.P17 (8)	GII.P17	Yes	GII.17 (6)	GII.17, GII.13
C4050	Yes	GII.P17 (7)	-	Yes	GII.17 (5)	-

NA: not applicable.

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
