# Peer review of "A Targeted Metagenomics Approach to Study the Diversity of Norovirus GII in Shellfish Implicated in Outbreaks"

_viruses, 2020, doi:10.3390/v12090978_

Round 1

Reviewer 1 Report

Major questions:

1. How were the shellfish samples selected for analysis? Please explain better the choice of shellfish samples. Did you include samples expected not to be contaminated? If not, such analysis should be done to reveal the sensitivity and specificity of the method.

2. How did you validate the reproducibility of the results? Should you analyze the same shellfish sample multiple times to demonstrate the reproducibility?

3. The composition of the artificial samples is difficult/impossible to grasp from the Table 2. What is the meaning of the percentage value provided in Table 2? Does NoV concentration refer to measured or theoretical value? Please explain thoroughly and improve the presentation. The Table 2 after revision should be easy to read and understand!

4. Lines 269-270: the authors claim that “the technique was able to clearly identify GII.P17 and GII.17 as the main P-type / genotype in these pools”. I partially disagree: For example, in sample B, the GII.17 was not found as main genotype in dilution 10 gc/ul.

5. Do we know if shellfish naturally carries norovirus-related viruses?

Minor comments:

6. Line 232: specify the negative control used.

7. Line 297: please check that the numbers provided are correct (987 appears very small number)

8. Figure 3 has poor quality.

Reviewer 2 Report

This paper conducted by Desdouits et al., describes a method to study NoV diversity in low-contaminated foods and the identification of NoV strains implicated in outbreaks. The work is very interesting, but it refers to strains circulating before 2016, it would be nice to make the work more current.

Considering only the applied method, it seems viable and also useful for the implementation of the epidemiology of Outbreaks of shellfish-associated 

Minor revision:

The bibliography of the introduction must be revised as it is obsolete. in particular line 36: obsolete bibliography. From 2103 onwards, NoVs have been linked more often to outbreaks transmitted by food such as berries and ready-to-eat salads than to outbreaks associated with the consumption of shellfish. Then edit that sentence.

Reviewer 3 Report

The authors performed metagenomics approach to examine the divergence of NoV GII in shellfishes implicated in past outbreaks. As a result, they showed that combining RdRp and VP1 metegenomic analyses was a useful approach for the shellfish-associated NoV outbreaks. Overall, the draft manuscript well designed and described. Methodology also was sound. Some minor issues should be addressed.

  1. All samples used in this study was harvested 4 years ago. Should describe the statement of the sample stock. Did they store at -80 C?
  2. Please carefully check as follows: space (ex. Line 77), insertion of "and" (ex. Line 52, 118), and footnote position (Line 311).             
